# Towards Energy-efficient Federated Learning via INT8-based Training on Mobile DSPs

## ABSTRACT

AI is making the Web an even cooler place, but also introduces serious privacy risks due to the extensive user data collection. Federated learning (FL), as a privacy-preserving machine learning paradigm, enables mobile devices to collaboratively learn a shared prediction model while keeping all training data on devices. However, a key obstacle towards practical cross-device FL training is huge energy consumption, especially for lightweight mobile devices.

In this work, we perform the first-of-its-kind analysis of improving FL performance through low-precision training with an energy-friendly Digital Signal Processor (DSP) on mobile devices. We first demonstrate that directly integrating the state-of-the-art INT8 (8-bit integer) training algorithm and classic FL protocols will significantly degrade the model accuracy. Moreover, we observe that there are still unavoidable frequent quantization operations on devices that cause extreme load stress on DSP-enabled INT8 training. To address the above challenges, we present `Q-FedUpdate`, an FL framework that efficiently preserves model accuracy with ultra-low energy consumption. It maintains a global full-precision model and allows the tiny model updates to be continuously accumulated, instead of being erased by the quantization. Furthermore, it introduces pipelining technology to parallel CPU-based quantization and DSP-enabled training, which reduces the floating-point computation overhead of frequent data quantization. Extensive experiments show that `Q-FedUpdate` can effectively reduce the on-device energy consumption by 21×, and accelerate the FL convergence by 6.1× with only 2% accuracy loss.

## CCS CONCEPTS

• **Human-centered computing** → *Ubiquitous and mobile computing systems and tools.*

## KEYWORDS

Mobile computing, federated learning, energy efficiency

## 1 INTRODUCTION

A colossal amount of data is being generated on Internet-of-Things or Web-of-Things devices, e.g., Web browsing traces and input corpora on smartphones. Harnessing such data is vital to the success of AI-driven Web systems [12, 29, 41], yet it must be done in a privacy-friendly manner as enforced by recent regulations [1, 2]. Federated learning (FL) [21] is by far one of the most effective approach that proposed to achieve privacy-preserving machine learning, and has already landed many killer mobile applications, such as Web browser history suggestions [31], input method prediction [14], and voice assistant [3].

However, a key obstacle towards practical cross-device FL training to serve mobile applications is the huge energy consumption.

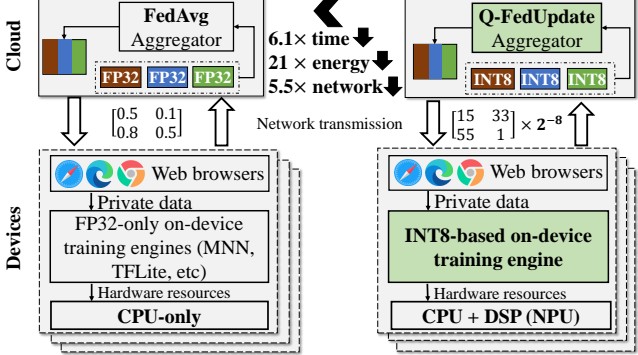

**(a) Traditional FP32-based FL**  **(b) Proposed INT8-based FL**

**Figure 1: A comparison of traditional FP32-based FL and our INT8-based FL (FP32: 32-bit float; INT8: 8-bit integer) across mobile devices. With the deployment of INT8 training on DSPs, it enables superior mobile Web services while keeping all sensitive data on devices.**

For example, our preliminary experiment of FL training with VGG-16 [35] on CIFAR-10 [23] consumes power 42.5 kJ for each participant device, which equals watching a YouTube 1080p video on a smartphone for about 24 hours. The root cause is twofold: (1) the end-to-end convergence typically takes hundreds or even thousands of global rounds due to the non-IID data distribution, e.g., 3,000 rounds for an RNN model reported by Google [7]; (2) per round, on-device training is also slow due to the constrained hardware resources, e.g., 2.1 seconds to train a VGG-16 model with just one batch (size 64).

To mitigate such energy impacts on user experience, existing FL deployments only consider devices that are idle and battery-charged [7]. This constraint makes FL less flexible and could harm the accuracy of the output model due to the client selection bias [24, 25]. Prior FL performance optimizations mostly focus on convergence speed and network communication cost [27, 30, 37]. Recently, the ML community has explored *low-precision training* [6, 13, 38, 42, 47, 48] to reduce the DNN training overhead, which leverages fewer-bit data format such as INT8 (8-bit integer) to represent weights/activations during training instead of FP32 (32-bit float). While these have proven superior performance on a single device, they lack key insights into end-to-end FL performance, such as convergence accuracy and related cost of energy and time.

In this work, we aim to make the cross-device FL more energy-efficient by incorporating low-precision training based on two observations. (1) Low-precision training can utilize more heterogeneous and energy-efficient processors, as integer operations are much more efficient than floating-point operations. Indeed, on mobile devices, FP32-based training can only be performed on CPUs,

even not on GPU [11, 39]. (2) Modern mobile devices often incorporate domain-specific processors that are powerful on integer-based operations. For instance, Digital Signal Processor (DSP) is ubiquitously available on smartphones and adequate to execute 128x INT8 operations in one cycle. It shows an 11.3×/4.0× improvement of energy saving over CPU/GPU in ML tasks [4]. To this end, we make the very early exploration on *orchestrating the low-precision training on mobile DSPs with FL protocols to improve the energy efficiency in cross-device FL scenario*. Its overall concept and the achievements made are illustrated in Figure 1.

However, directly applying the state-of-the-art INT8 training algorithms [38] into cross-device FL framework with mobile DSP deployment still faces two critical challenges. First, it exhibits significant performance degradation when combining these algorithms and FL protocols such as *FedAvg* [30]. For example, from our preliminary experiment on CIFAR-10 and VGG-16, we observe 16% accuracy loss and 1.25× slower convergence compared with traditional FP32-based FL. We find out the reason that, during the aggregation, most of the values in *model updates* generated by local INT8 training are even smaller than the numerical error of INT8 quantization. Those updates are therefore erased when the new global FP32 model is quantized back to INT8 to be dispatched to devices. Second, pure DSP-driven INT8 training cannot efficiently run on devices, due to the inevitable FP32 operations introduced by frequent batch-data quantization.

Atop those key observations, we propose Q-FedUpdate, an INT8 FL framework that significantly reduces energy consumption of mobile devices and accelerates model convergence. The key idea of Q-FedUpdate is twofold. First, it designs an error-compensated aggregation protocol that aggregates the *model updates* produced by the local INT8 training, instead of the locally-updated model weights. And the aggregated result is applied on a maintained global FP32 model to compensate for the quantization errors before proceeding to the next round. The rationale is that it can preserve more *effective updates* that will not be erased by the by the FP32-to-INT8 quantization, though aggregating model updates is mathematically equivalent to model weights aggregation with FP32-only format. In such a way, the tiny *model updates* are accumulated during the FL process and finally take effect in the quantized model. Second, our key observation is that CPU-based data quantization and DSP-enabled model training are not strictly dependent, as there is non-negligible PCIe transmission in between. it designs a pipelined batch quantization mechanism, which pipelines data quantization on CPU, transmission across PCIe and INT8 training on DSP. It enables an efficient parallelization of CPU-based quantization and DSP-enabled training to alleviat DSP's computation pressure.

We have implemented Q-FedUpdate on a simulation platform atop FLASH [43] and an end-to-end INT8 training library for smartphones based on MNN [18], a popular lightweight ML engine developed by Alibaba. We then perform extensive experiments to evaluate the effectiveness of Q-FedUpdate on mobile CPU/DSP chips with 3 classical FL datasets. The results show that, compared to traditional FP32-based FL protocol, Q-FedUpdate tremendously reduces energy consumption of mobile devices by 14×-28×, and significantly accelerates the model convergence by 5.1×-7.1×, with acceptable accuracy loss (1%-3%).

Contributions are summarized as follows.

**Table 1: The inference time (T: ms) and energy consumption (E: J) of different model precision. The experiments are performed on Xiaomi 10 with TensorFlow Lite.**

| Models | CPU, FP32 | | CPU, INT8 | | DSP, INT8 | |
|---|---|---|---|---|---|---|
| | T | E | T | E | T | E |
| MobileNet-V1 | 11.4 | 88.2 | 4.3 | 22.7 | 2.5 | 5.4 |
| MobileNet-V2 | 8.6 | 64.8 | 4.6 | 22.9 | 3.1 | 5.4 |
| ResNet-50 | 78.7 | 597.6 | 27.8 | 131.4 | 9.2 | 28.8 |
| Inception-V4 | 266.3 | 1,980 | 81.5 | 399.6 | 17.2 | 59.4 |
| EfficientNet-V2 | 33.0 | 187.2 | 13.4 | 59.4 | 8.44 | 12.6 |

- To our best knowledge, we are the first to explore the idea of applying DSP-enabled INT8 training on FL, and show that traditional FL protocol does not fit INT8 training through both experimental and theoretical analysis.
- We propose Q-FedUpdate, a energy-efficient FL framework for privacy-preserving multimedia training task. It incorporates two key techniques: error-compensated aggregation protocol and pipelined batch quantization mechanism.
- We implement Q-FedUpdate through simulation and an on-device training library. Extensive experiments show the effectiveness of Q-FedUpdate compared to competitive baselines.

## 2 BACKGROUND AND MOTIVATIONS

### 2.1 Definition of Federated Learning

Federated learning (FL) is a popular distributed learning paradigm where $N$ clients owning heterogeneous data distributions collaboratively learn a global model $w$. A typical FL process incorporates a central server to aggregate weights from clients [21]. Existing FL protocols, e.g., *FedAvg* [30], mostly adopt stochastic gradient descend (SGD) to optimize the local loss function $f$ with $E$ epochs and a fixed learning rate $\eta$. In particular, only $K$ devices are randomly selected form total $N$ devices at each global training round $t$. The cloud then aggregates the updates sent by each device $k$ across WAN. The updates can be formulated as $w^k(t+1) \leftarrow w^k(t) - \eta \nabla f_k(w^k(t))$ and then $w(t+1) \leftarrow \sum_{k=1}^{K} \frac{n_k}{n} \cdot w^k(t+1)$.

### 2.2 DSP-enabled INT8 Training

DSP was originally designed for processing digital signals like audio with high energy efficiency. The Hexagon 680 DSP with Hexagon vector extensions, announced by Qualcomm in 2016, firstly allow significant compute workloads for advanced image processing and computer vision [4]. This DSP contains Hexagon cores and a single-instruction-multiple-data co-processor, which is good at vector computation [7]. It can process 1024-bit fixed-point data inside one instruction, or 128 INT8 mathematical functions like add and multiply in one cycle. Besides, the Hexagon core's clock frequency is 500 MHz, which is much lower than the CPU ones, so it is much more energy friendly. We have measured the speed and energy of 5 typical deep learning models on Xiaomi 10, and summarized the results in Table 1. It shows that the execution speed is accelerated by 2.8×-15.5× on DSP. The energy saving on DSP is even more profound: since DSP runs at low frequency, it reduces energy consumption by up to 32×.

**Algorithm 1:** *Quantized Federated Averaging (*Q-FedAvg*)*

---

**Input** : selected clients $K$, local epochs $E$, learning rate $\eta$,
         initialized global FP32 model $w(0)$.
**Output**: the global FP32 model $w$.

1  ***Server executes:*** // $w_q$: INT8 model quantized from global
    FP32 model $w$, $w_d$: FP32 model dequantized from $w_q$.
2     **for** $t \leftarrow 0$ **to** $T$ **do**
3         $w_q(t) \leftarrow$ quantize $w(t)$ // Quantization
4         $S(t) \leftarrow$ (random set of $K$ selected clients)
5         **for** *each client* $k \in S(t)$ *in parallel* **do**
6             $w_q^k(t+1) \leftarrow$ **ClientUpdate**$(k, w_q(t))$
7             $w_d^k(t+1) \leftarrow$ dequantize $w_q^k(t+1)$
8         $w(t+1) \leftarrow \sum_{k=1}^{K} \frac{n_k}{n} w_d^k(t+1)$ // global FP32 model
9  ***ClientUpdate(**k, w_q**):** // local training on client $k$
10    **for** *each epoch* $i \leftarrow 1$ **to** $E$ **do**
11        **for** *each training data batch* $b$ **do**
12            $b_q \leftarrow$ quantize $b$ // Quantization
13            $g \leftarrow \nabla f(w_q; b_q)$ // $g$ is stored as INT32 format
14            $g_\eta \leftarrow round_s(g \gg (\log_2 B(g) - \eta))$ // Eq.2
15            $w_q \leftarrow w_q - g_\eta$ // update local INT8 model
16    **return** $w_q$ to server

---

To maximize the advantage of integer-only arithmetic operation in DSP, we propose to introduce the popular low-bit DNN training algorithm [24]. In the following, we describe the INT8 (8-bit integer) training process enabled by DSP from three aspects: quantization, forward and backward propagation. We mainly retrofit the INT8 training algorithm from a state-of-the-art work [38] for its nearly accuracy lossless and integer-only arithmetic.

**Quantize and Dequantize**. For each FP32 numerical value in the initialized weights or input batches denoted as $x$, we follow the symmetric uniform quantization [22] to quantize it to an INT8 number. This quantization method is the most efficient quantization scheme due to its hardware-friendly computation. For each $x$ following in the range $(l, u)$ and a clipping value $c \in (0, max(|l|, |u|))$, the quantization can be formulated as below:

$$x_q = round(\frac{clip(x, c)}{s}), \tag{1}$$

where $clip(x, c) = min(max(x, -c), c)$, $s = \frac{c}{2^{(8-1)}-1}$ indicates the scaling factor, and $x_q$ represents the quantized INT8 number. The round operation here is a conventional rounding-off method. Subsequently, the corresponding dequantized data $x_d = x_q \cdot s$. Noting that the dequantized FP32 $x_d$ has an inevitable gap between the original input FP32 $x$.

**Forward pass**. After the above quantization operations, the layer input $X$ and weight $W$ are INT8 numbers. The activations are denoted as $A$ to store the INT32 numbers, which are transformed to INT8 numbers by the right shift and stochastic rounding operations [13]. All multiply and shift&round operations are executed on DSP in lower frequency. When forwarding to the final layer, the activations are rounded to INT8 output $Y$. With the integer cross-entropy loss arithmetic designed in [38], we can obtain the INT32 error gradient $g_Y$.

**Backward pass**. The obtained INT32 error gradients $g_Y$ are rounded to INT8 gradients of the last layer denoted as $g_{A^l}$. Then we recursively calculate each layer $l$'s gradient $g_{A^l}$ with respect to this

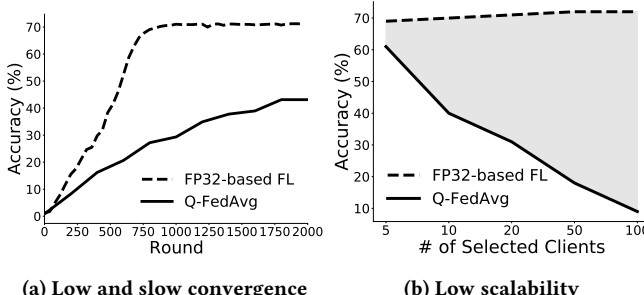

**(a) Low and slow convergence**      **(b) Low scalability**

**Figure 2: `Q-FedAvg`'s performance degradation as compared to traditional FP32-based FL (CIFAR-100). Notice that the y-axis in (b) denotes the final convergence accuracy.**

layer's activations $A^l$. The INT32 activations and gradients are rounded into INT8 values, similar to the forward pass, before being back-propagated to the previous layer. Each layer needs to derive two sets of gradients to perform the recursive update. (1) The layer activation gradients, $g_{A^{(l-1)}} = g_{A^l} \cdot W^l$, which is passed to the next layer. (2) The weight gradients, $g_{W^l} = g_{A^l} \cdot A^{(l-1)}$, which is used to update layer $l$'s weights $W^l$.

Next, the SGD algorithm is adopted to update the current layer's weight $W^l$ with the calculated weight gradients $g_{W^l}$. The FP32-based SGD is formulated as $W^l = W^l - \eta \cdot g_{W^l}$, where $\eta$ denotes the FP32 learning rate. However, the FP32-based multiplication is not allowed in INT8 training. Therefore, we follow the previous work [38] to replace the $\eta \cdot g_{W^l}$ with the following operations:

$$g_\eta^l = round_s(g_{W^l} \gg (\log_2 B(g_{W^l}) - \eta)), \tag{2}$$

where $\eta$ denotes the INT8 learning rate, $\gg$ denotes the right shift operation, and $\log_2 B(g_{W^l})$ denotes the bit width that can represent each INT32 number in $g_{W^l}$. Therefore, the INT8 $\eta$ in Eq.2 essentially represents the final bit width of $g_\eta^l$ after the shift and rounding.

### 2.3 Motivations

In this work, we aim to design an INT8 FL protocol, which utilizes DSP-enabled INT8 training to accelerate the model convergence and reduce the power consumption during the FL process. However, our preliminary experiments highlight the insufficiency of directly applying DSP-enabled INT8 training in FL. The observed insufficiency is two-fold: 1, Popular FedAvg algorithm severely decreases the convergence performance; 2, DSP is not friendly for all operators (especially quantization) during on-device INT8 training.

**Directly applying INT8 training to FedAvg incurs low precision and scalability.** The most intuitive way to achieve quantized federated learning is to directly combine INT8 training with *FedAvg* [30], the most widely adopted algorithm in FL. We name this approach as `Q-FedAvg` (details in Algorithm 1). Our experiments show that `Q-FedAvg` exhibits significant performance degradation as compared to the traditional FP32-based training. The settings of the experiments are consistent with §4.

(1) `Q-FedAvg` results in nontrivial accuracy degradation and requires more rounds to converge. As shown in Figure 2(a), the convergence accuracy of `Q-FedAvg` is 30% lower than FP32-based FL. It also takes 1.5× more global rounds towards model convergence.

Consequently, Q-FedAvg not only decreases the model accuracy, but also fails to achieve convergence acceleration as expected. (2) FedAvg is known to benefit from more clients involved in per-round training [24], i.e., higher and more stable convergence accuracy. In Q-FedAvg, however, we observe that more clients lead to lower accuracy. As shown in Figure 2(b), when the selected client number increases from 5 to 100, the accuracy drops by 62% in Q-FedAvg. Such low scalability indicates that Q-FedAvg can only utilize a small portion of available devices simultaneously. The intuitive behind reason of above insufficiency is that the numerical error inevitably introduced by low precision representation will aggravate the bias between device models, especially with the Non-IID data distributions across devices [30].

**DSP is poorly suited for performing quantization operations with FP32 arithmetic.** Although the DSP is great for handling INT8 vector arithmetic operations, there are still numerous FP32 operations involved in batch data quantization (as shown in Line 12 of Algorithm 1) that cannot be efficiently executed on a DSP. We conducted preliminary measurements to compare the performance of FP32 operations on the CPU and DSP. The computation time for FP32-based quantization on the DSP is as high as 21.3 ms, respectively, which is nearly 9× more than that on the CPU. This observation has led us to develop an efficient coordination strategy between the CPU and DSP.

## 3 Q-FEDUPDATE: AN INT8 FL FRAMEWORK

In this section, we propose quantized federated update (namely Q-FedUpdate), an energy-efficient FL framework incorporated with two key techniques to address the above challenges. Algorithm 2 elaborates their efficient design on server and client, respectively. We first analyze the essential reasons to the performance deficiencies of Q-FedAvg, which guides us to design an error-compensated aggregation protocol on server, as shown in §3.1. Then, we propose a pipelined batch quantization mechanism during the on-device INT8 training to alleviat the heavy computation overhead of FP32-based quantization and dequantization in §3.2.

### 3.1 Error-Compensated Aggregation

To further investigate the essential reasons behind such a phenomenon, we first conduct a detailed analysis on how the model weight values change during the whole Q-FedAvg training. The key reason is that most of the aggregated model updates (Line 8) are too small compared to the model weights, so that they are mostly erased during quantization (Line 3) after being applied to the weights in the next round. In other words, most weights are not intrinsically updated when moving on to the next round of INT8 training. Atop this observation, we then design the error-compensated aggregation protocol that applies these tiny aggregated model updates on a maintained global FP32 model.

**Why is Q-FedAvg insufficient?** We first define the *model updates* of client $k$ at round $t + 1$ as:

$$\Delta w_d^k(t + 1) = w_d(t) - w_d^k(t + 1), \tag{3}$$

where $w_d(t)$ denotes the dequantized FP32 model from global INT8 model $w_q(t)$ at round $t$. It essentially represents the accumulated model updates after several local epoch INT8 training on client $k$.

---

**Algorithm 2:** *Quantized Federated Update (Q-FedUpdate).*

---

**Input** : selected clients $K$, initialized global FP32 model $w_0$.
**Output**: the global FP32 model $w$.
   // Error-compensated aggregation on server
1  ***Server executes:*** // $w_q$: INT8 model quantized from global
    FP32 model $w$, $w_d$: FP32 model dequantized from $w_q$.
2    **for** $t \leftarrow 0$ **to** $T$ **do**
3       $w_q(t) \leftarrow$ quantize $w(t)$ // Quantization
4       $w_d(t) \leftarrow$ dequantize $w_q(t)$ // Dequantization
5       $S(t) \leftarrow$ (random set of $K$ selected clients)
6       **for** *each client* $k \in S(t)$ *in parallel* **do**
7          $w_q^k(t + 1) \leftarrow$ **ClientUpdate**$(k, w_d(t))$
8          $w_d^k(t + 1) \leftarrow$ dequantize $w_q^k(t + 1)$
9          $\Delta w_d^k(t + 1) \leftarrow w_d(t) - w_d^k(t + 1)$ // model updates
10       $\Delta w_d(t + 1) \leftarrow \sum_{k=1}^{K} \frac{n_k}{n} \Delta w_d^k(t + 1)$ // global updates
11       $w(t + 1) \leftarrow w(t) - \Delta w_d(t + 1)$ // global FP32 model
   // Pipelined batch quantization during local training
12  ***ClientUpdate(k, $w_q$):*** // local training on client $k$
13    **for** *each epoch* $i \leftarrow 1$ **to** $E$ **do**
14       **for** *each training data batch* **in pipeline mode** **do**
         // Pipelined batch quantization
15          $P_1$: $b \leftarrow$ sample each batch
16          $P_2$: $b_q \leftarrow$ quantize $b$
17          $P_3$: transmit $b_q$ across PCIe
18          $P_4$: $g \leftarrow \nabla f(w_q; b_q)$; $g_\eta \leftarrow round_s(g \gg (\log_2 B(g) - \eta))$; $w_q \leftarrow w_q - g_\eta$
19    **return** $w_q$ to server

---

The global *model updates* at round $t + 1$ can be calculated based on Eq.3 as follows:

$$\Delta w_d(t + 1) = \sum_{k=1}^{K} \frac{n_k}{n} (w_d(t) - w_d^k(t + 1)). \tag{4}$$

Then, we have $\Delta w_d(t + 1) = w_d(t) - \sum_{k=1}^{K} \frac{n_k}{n} \cdot w_d^k(t + 1)$, due to $\sum_{k=1}^{K} \frac{n_k}{n} \cdot w_d(t) = w_d(t)$. Compared to the global FP32 model aggregation (Line 8) of Q-FedAvg, we find it equivalent to $w(t + 1) \leftarrow w_d(t) - \Delta w_d(t + 1)$. We then detailly analyze the impact of quantization on this global FP32 model. At the beginning of each round, the updated global FP32 model $w(t + 1)$ is quantized to INT8 model as Eq. 1:

$$w_q(t + 1) = round(\frac{clip((w(t + 1)), c)}{s})$$
$$= round(\frac{clip((w_d(t) - \Delta w_d(t + 1)), c)}{s}). \tag{5}$$

Each weight value $x$ in $w_d(t)$ is represented numerically as $\alpha \cdot 2^\beta$. Due to FP32 model normalization at the beginning of model initialization, the value $x$ tends to satisfy $-c \leq x \leq c$, thus $clip(x, c) = x$. And we have $s = \frac{c}{2^{(8-1)} - 1}$ as in Eq 1. So, the above equation is equivalent to:

$$w_q(t + 1) = round(\frac{\alpha \cdot 2^\beta - \Delta w_d(t + 1)}{c} \cdot (2^{(8-1)} - 1)), \tag{6}$$

where $c = (2^{(8-1)} - 1) \cdot 2^\beta$ in our training process. Thus, we have:

$$w_q(t + 1) = round(\alpha - \frac{\Delta w_d(t + 1)}{2^\beta}). \tag{7}$$

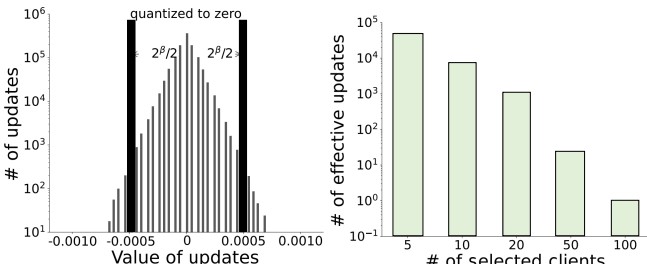

**(a) Distribution of *model updates*  (b) *Effective updates* comparison**

**Figure 3: Numerical analysis for `Q-FedAvg`'s performance degradation (CIFAR-100). Note that y-axis is logarithmic.**

Considering the rounding mechnism of the function round, if the term $\frac{\Delta w_d(t+1)}{2^\beta} < 0.5$, the INT8 value of $w_q(t+1)$ is still $\alpha$. Therefore, only the weight update values in $\Delta w_d(t+1)$ that exceed $\frac{2^\beta}{2}$ can be added to the global INT8 model in the next round, without being erased by quantization. We define such weight updates as *effective updates*, which is a key insight for our design of `Q-FedUpdate`.

We then conduct concrete experiments on the numerical distribution of $\Delta w_d$ during `Q-FedAvg` training. We surprisingly obtain the following unique observations: as shown in Figure 3(a), only a tiny portion (0.1%) of $\Delta w_d$ exceed $\frac{2^\beta}{2}$, while others will be quantized to zero after being applied to the original weights. It indicates that the efforts of multi-epoch training on local devices are wasted, and explains why `Q-FedAvg`'s performance degrades significantly.

To further explain the decreased scalability of `Q-FedAvg`, we calculate the number of effective updates with different numbers of clients selected. As shown in Figure 3(b), the number of effective updates decreases when the number of selected clients per round increases. We then explain the reason behind this phenomenon. As we analyze above in Figure 3(a), each selected client has a number of weight updates quantized to zero. Due to the weighted average operation, these zero updates in some clients affect the *effective updates* in some other clients, thus further reducing the total *effective updates* of the final aggregated model.

**How to compensate for this quantization error?** Motivated by the above analysis, our proposed error-compensated aggregation protocol updates the global FP32 model with the *model updates*, instead of the global INT8 model. This novel design compensates for the quantization error in this round with a maintained global FP32 model, which enables more *effective updates* for the quantized global INT8 model in the next round. Algorithm 2 shows its workflow on server. Compared to naive `Q-FedAvg`, there are several key steps: i) dequantize the quantized INT8 model at the beginning of each round (Line 4), ii) calculate and aggregate model updates (Lines 9-10), iii) update the global maintained FP32 model with the aggregated model updates (Line 11).

The design of this protocol is somehow counterintuitive as it applies the *model updates* on the global FP32 model, which is not the starting point of the model that is trained on each local device. We then explain why this design is more efficient and scalable compared to `Q-FedAvg`. As shown in Algorithm 1 and Algorithm 2, they both have the same initialized global FP32 model and global INT8 model. However, they gradually deviated after several global

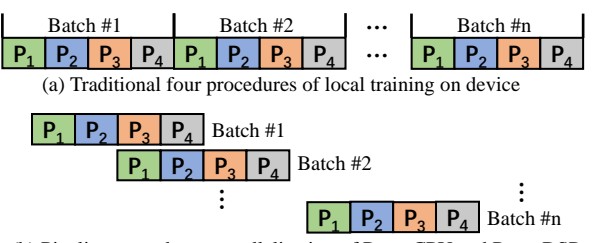

**(a) Traditional four procedures of local training on device**

P1 P2 P3 P4 Batch #1

P1 P2 P3 P4 Batch #2

P1 P2 P3 P4 Batch #n

**(b) Pipeline procedures: parallelization of P₂ on CPU and P₄ on DSP**

**Figure 4: `Q-FedUpdate` pipelines four preced­ures in local training: sample and quantize batch data ($P_1$&$P_2$ on CPU), transmit quantized data ($P_3$ across PCIe), and INT8-based forward and backward pass ($P_4$ on DSP).**

rounds. For this protocol in `Q-FedUpdate`,

$$w(t+1) = w(t) - \sum_{k=1}^{K} \frac{n_k}{n} \Delta w_d^k(t+1) \tag{8}$$

$$= w(t) - \sum_{k=1}^{K} \frac{n_k}{n}(w_d(t) - w_d^k(t+1))$$

$$= \underline{w(t) - w_d(t)} + \sum_{k=1}^{K} \frac{n_k}{n} w_d^k(t+1).$$

And for `Q-FedAvg`, $w_{t+1} = \sum_{k=1}^{K} \frac{n_k}{n} w_d^k(t+1)$. we observe from the comparison that `Q-FedUpdate` essentially compensates for the quantization error $w(t) - w_d(t)$ of the global model at each round $t$. Following the analysis of *model updates* in §3.2, this value can compensate the gap between the $\Delta w_d(t+1)$ and $\frac{2^\beta}{2}$ in Eq. 7. Therefore, it can recover a lot of weight information lost due to the quantization, thus improving the model performance.

## 3.2 Pipelined Batch Quantization

This section presents a pipelined batch quantization mechanism on device, as shown in Algorithm 2. Given the DSP-unfriendly on the quantization of batch data (illustrated in §2.3), we extend this operation on the CPU. As a result, the local training in `Q-FedAvg` becomes the following four procedures:

- **Sample a batch of training data on CPU (P1);**
- **Quantize batch data on CPU (P2);**
- **Transmit quantized data across PCIe (P3);**
- **INT8-based forward and backward pass on DSP (P4).**

However, since the CPU-DSP context switching incurs high overhead mainly due to the data copy between their own memory space (i.e., around 25ms on XiaoMI 10), this approach could lead to non-optimal performance.

Our key observation is that the training data quantization and model training in two successive batches are independent. In this case, the quantization of next batch does not need to wait for the DSP-enabled model training on the previous batch. Instead, we can start the CPU computation to quantize the next batch as soon as the previous quantized batch begin to be transmitted across PCIe, regardless of its following local training on DSP. Figure 4 illustrates the advantage of pipelining over the strawman solution.

**Table 2: The datasets and models used in experiments, and our measured per-batch training time (T: ms) and energy consumption (E: J) of those models on Xiaomi 10 with on-device training engine MNN [18].**

| Dataset | Model (size) | Batch size | # of clients | # of samples on each client | CPU, FP32 | | CPU, INT8 | | DSP, INT8 | |
|---|---|---|---|---|---|---|---|---|---|---|
| | | | | | T (ms) | E (J) | T (ms) | E (J) | T (ms) | E (J) |
| FEMNIST | LENET-5 (1.6MB) | 5 | 192 | 226 | 58 | 0.3 | 30 | 0.15 | 11 | 0.02 |
| CIFAR-10 | VGG-16 (15MB) | 64 | 100 | 500 | 2,076 | 13.6 | 1,075 | 6.9 | 397 | 1.1 |
| CIFAR-100 | VGG-16 (15MB) | 64 | 500 | 100 | 2,096 | 14 | 1,080 | 7.2 | 401 | 1.1 |

**Table 3: Our measured per-batch time and energy as Table 2 on three devices with different levels of DSP chips.**

| Device | DSP level | FEMNIST | | CIFAR-10 | | CIFAR-100 | |
|---|---|---|---|---|---|---|---|
| | | T | E | T | E | T | E |
| Redmi Note 9 | Low | 23 | 0.03 | 810 | 1.7 | 812 | 1.7 |
| Xiaomi 10 | Medium | 11 | 0.02 | 397 | 1.1 | 401 | 1.1 |
| Xiaomi 11 Pro | High | 8 | 0.02 | 300 | 1 | 309 | 1 |

## 4 EVALUATION

### 4.1 Experiment Settings

**Datasets and models.** In Table 2, we evaluate Q-FedUpdate on three real-world federated datasets and two models: LeNet-5 [10] on FEMNIST [32], with 192 clients as in previous work [8]. VGG-16 [35] on both CIFAR-10 [23] and CIFAR-100 [23], with each partitioned into 100 and 500 clients as in previous work [15].

**Simulation platform.** We have implemented Q-FedUpdate on a simulation platform atop FLASH [43] and an end-to-end INT8 training library for smartphones based on MNN [18], a lightweight DL engine developed by Alibaba. The platform also incorporates the concept of device heterogeneity as introduced by FLASH [43], which includes large-scale smartphone traces to simulate how FL operates once deployed in the real world. To obtain the on-device training performance with FP32/INT8 format on CPU/DSP, we follow prior work [42] to measure the wall clock time and energy consumption with MNN library on three different levels of smartphones. Overall, the per-batch training cost on Xiaomi 10 is summarized in Table 2 and the results on different DSP chips are shown in Table 3. We also plug the numbers measured into the simulation platform to obtain the end-to-end performance till the model convergence. All our simulated experiments are performed on a high-performance Linux server with 8 NVIDIA V100 GPUs.

**Metrics.** Apart from the *convergence accuracy* on the testing data, we also report the following two metrics that closely relate to the mobile devices that participate in FL. (1) *Energy* measures the average energy consumption of on-device training for each device, while considering the random device selection [30]. (2) *Clock time* is the end-to-end training time perceived by the FL developers, including many rounds of on-device training and network communication time to download/upload models with a default bandwidth (10Mbps on average) capacity [46].

**Baselines.** We compare Q-FedUpdate with three baselines: (1) *FloatFL*: the traditional FP32-based FL protocol with *FedAvg*. (2) Q-FedAvg: directly integrating INT8 training with *FedAvg* as shown in Algorithm 1. (3) *QuanFL* [40]: reducing the network transmission time through model quantization as INT8 format, but still using FP32-based training on devices.

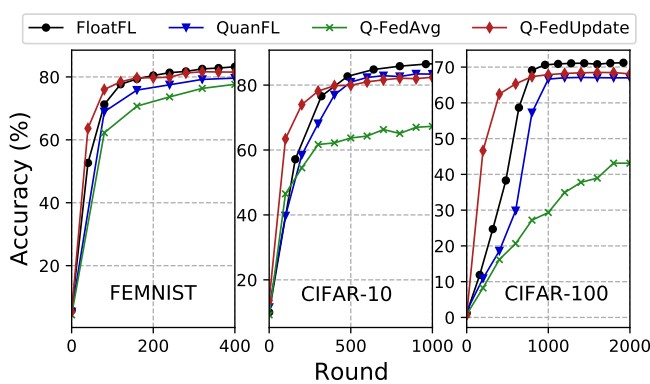

**Figure 5: The convergence accuracy of Q-FedUpdate and baselines across training round on three datasets.**

### 4.2 Convergence Analysis

In this section, we demonstrate that Q-FedUpdate is able to significantly accelerate the model convergence while guaranteeing the model accuracy. The results are illustrated in Figure 5. To obtain the highest model accuracy, we give the settings of the following hyperparameters. We follow the prior work [8, 38] to set $\eta$=3 for Q-FedUpdate and Q-FedAvg, and set $\eta = 0.01$ for FloatFL and QuanFL, respectively. We set $E$=1 for all FL protocols. We set $k$=5 for Q-FedAvg (fewer clients attributes to higher accuracy), and $k$=50 for others on three datasets.

**Q-FedUpdate vs. FloatFL and Q-FedAvg.** As shown in Figure 5, compared with FloatFL, Q-FedUpdate greatly improves the convergence speed while preserving the model accuracy. For instance, on FEMNIST, Q-FedUpdate's convergence accuracy is only 1% lower than FloatFL, but it takes 35.0% fewer global rounds to converge. Similarly, on CIFAR-10 and CIFAR-100, Q-FedUpdate's convergence accuracy is 3% and 2% lower than FloatFL, but it takes 12.5% and 20.0% fewer global rounds to converge, respectively. The 2% average accuracy loss is mainly due to the INT8 format, which has limited numerical representation. This accuracy drop is generally accepted by the relevant low-precision training community [6, 13, 38, 47, 48]. At the same time, INT8-based training is also the key reason why Q-FedUpdate can achieve such a speedup. Compared to Q-FedAvg, Q-FedUpdate achieves great accuracy improvement with much fewer global rounds. More specifically, Q-FedUpdate achieves 3%, 13%, and 8% higher convergence accuracy, and greatly reduces 43.5%, 80.0%, and 30.0% global rounds required to converge on the three datasets. Such tremendous improvement comes from our unique design of Algorithm 2 (in §3.1) that updating a maintained global FP32 model with the *model updates*, instead of the updated INT8 model.

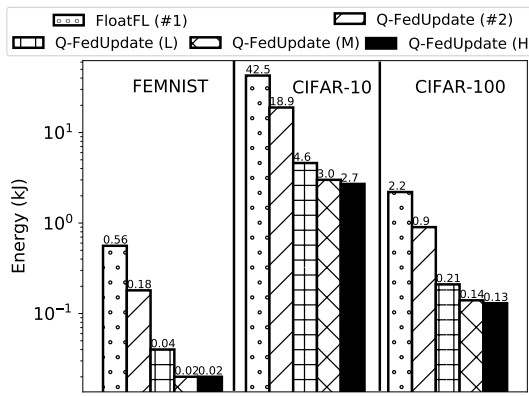

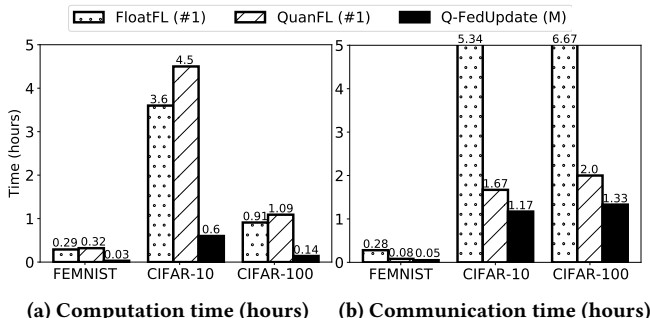

(a) Computation time (hours)  (b) Communication time (hours)

**Figure 7: The acceleration performance breakdown of computation and communication time.**

**Figure 6: Impact analysis of `Q-FedUpdate` using different DSP chips compared with FloatFL (#1). Breakdown analysis of `Q-FedUpdate` design and DSP hardware compared with CPU-enabled FloatFL (#1) and `Q-FedUpdate` (#2).**

**`Q-FedUpdate` vs. QuanFL.** The accuracy of `Q-FedUpdate` on the three datasets is nearly the same as that of QuanFL. QuanFL transmits the quantized INT8 models to the server, but still uses FP32-based training on devices. Therefore, the communication cost of `Q-FedUpdate` is the same as that of QuanFL in each round. But, it takes 40.9%, 30.0%, and 33.3% fewer global rounds for `Q-FedUpdate` to converge, due to the higher efficiency of INT8 training [47].

## 4.3 Energy and Time Reduction

In this section, we analyze the energy and time consumed for `Q-FedUpdate` to converge in an end-to-end manner. These two metrics are highly concerned by mobile devices. `Q-FedUpdate` is tested on both CPU and 3 different DSP chips in Table 2, 3.

**Energy-to-converge.** As observed from Table 4, `Q-FedUpdate` takes only 0.02, 3.0 and 2.6 (unit: kJ) energy consumption per client using medium DSP to reach convergence on FEMNIST, CIFAR-10 and CIFAR-100, respectively. It exhibits 14×–28× energy reduction as compared to traditional FloatFL, with only around 2% accuracy loss. Compared with a stronger baseline QuanFL, `Q-FedUpdate` still reduces 3.2× energy consumption on average. This is because, QuanFL only optimizes for network traffic but not on-device computation. Such an improvement on energy efficiency makes it more flexible to select clients that may not have abound batteries.

To investigate the impact of different DSP chips on energy consumption, we compare `Q-FedUpdate` using low/medium/high DSPs to the traditional FloatFL using CPU as shown in Figure 6. The energy consumption of `Q-FedUpdate` increases as the performance of DSP decreases. For example, `Q-FedUpdate` using a low DSP consumes up to 70.4% more energy than that using a high DSP. Compared with FloatFL on CPU, even deploying `Q-FedUpdate` on wimpy DSP can bring 9×-14× energy reduction (details in Table 4).

We also deploy the INT8 training on CPU to analyze the breakdown improvement of energy consumption due to the `Q-FedUpdate` design and DSP hardware. We compare `Q-FedUpdate` using medium DSP, `Q-FedUpdate` using CPU and FloatFL as shown in Figure 6. Compared with traditional FloatFL, `Q-FedUpdate` using CPU still reduces energy by up to 2.2×-3.1× on the three datasets. This is

because CPU is better at integer operations (though not as good as DSP) compared to floating points. Further enabling DSP with `Q-FedUpdate` reduces energy by 6.3×-9.0× for a medium DSP.

**Time-to-converge.** As shown in Table 4, it takes only 0.08, 1.75 and 1.47 hours for `Q-FedUpdate` using medium DSP to converge on FEMNIST, CIFAR-10 and CIFAR-100, respectively. It is 5.1×-7.1× faster compared to FloatFL. Such tremendous improvement comes from both on-device training acceleration and network traffic reduction. On the other hand, QuanFL also reduces the network transmission time through model quantization, yet it is still 2.1×-5.0× slower than `Q-FedUpdate`. We then break down the clock time to delve into the improvement on computation (on-device training) and communication time separately, in Figure 7. We obtain two key observations from this figure: (1) `Q-FedUpdate` reduces both the computation and communication time using DSP, while QuanFL only reduces communication time, and even increases computation time (1.1×-1.3×) due to the slower convergence. (2) `Q-FedUpdate` achieves 7.9× average computation acceleration, which is far more significant than its 5.1× average communication acceleration.

Similarly, Table 4 also shows the acceleration performance of `Q-FedUpdate` using different DSP chips. First, the convergence time of `Q-FedUpdate` increases as DSP's performance decreases (up to 71.4% on FEMNIST). Second, even `Q-FedUpdate` using wimpy DSP converges faster than all other baselines, including `Q-FedUpdate` using CPU. And it still accelerates the convergence by up to 3.7×-4.8× compared with traditional FloatFL.

Finally, we compare traditional FloatFL, `Q-FedUpdate` using CPU and `Q-FedUpdate` using medium DSP to analyze the breakdown improvement of convergence acceleration as shown in Table 4. With INT8 training deployed on medium DSP instead of CPU, `Q-FedUpdate` accelerates the convergence by up to and 1.2×-1.8× on the three datasets . These benefits come from the advantages of DSP hardware for integer operations. In addition, `Q-FedUpdate` using CPU further accelerates the convergence by up to 3.2×-4.1× in terms of hours, compared with FloatFL using CPUs. This huge acceleration comes from the weight compensation design of `Q-FedUpdate`, which enables fewer global rounds to converge.

## 4.4 Scalability Analysis on Selected Clients

In this section, we study the scalability of `Q-FedUpdate` with different number of selected clients ($K$=5,10,20,50,100) per round.

**Table 4: The convergence accuracy (Acc: %) and the time (T: hours) / energy consumed (averaged per client, E: kJ) to reach that convergence. We denote the settings as (#1):(CPU,FP32), (#2):(CPU,INT8), which means FP32/INT8 training on CPU processor as shown in Table 2. And (L/M/H) represents (L-DSP/M-DSP/H-DSP,INT8), which means INT8 training on low/medium/high DSPs as shown in Table 3, respectively.**

| Algorithms | FEMNIST | | | CIFAR-10 | | | CIFAR-100 | | |
|---|---|---|---|---|---|---|---|---|---|
| | Acc (%) | T (hours) | E (kJ) | Acc (%) | T (hours) | E (kJ) | Acc (%) | T (hours) | E (kJ) |
| FloatFL (#1) | **82 (1↓)** | 0.57 | 0.56 | **84 (3↓)** | 8.94 | 42.5 | **71 (2↓)** | 7.58 | 2.2 |
| QuanFL (#1) | 80 | 0.40 | 0.62 | 82 | 6.17 | 53.1 | 67 | 3.09 | 2.6 |
| Q-FedUpdate (#2) | 81 | 0.14 | 0.18 | 81 | 2.80 | 18.9 | 69 | 1.71 | 0.9 |
| Q-FedUpdate (L) | 81 | **0.12 (4.8×)** | **0.04 (14×)** | 81 | **2.40 (3.7×)** | **4.6 (9×)** | 69 | **1.61 (4.7×)** | **0.21 (10×)** |
| Q-FedUpdate (M) | 81 | **0.08 (7.1×)** | **0.02 (28×)** | 81 | **1.77 (5.1×)** | **3.0 (14×)** | 69 | **1.47 (5.2×)** | **0.14 (16×)** |
| Q-FedUpdate (H) | 81 | **0.07 (8.1×)** | **0.02 (28×)** | 81 | **1.62 (5.5×)** | **2.7 (16×)** | 69 | **1.44 (5.3×)** | **0.13 (17×)** |

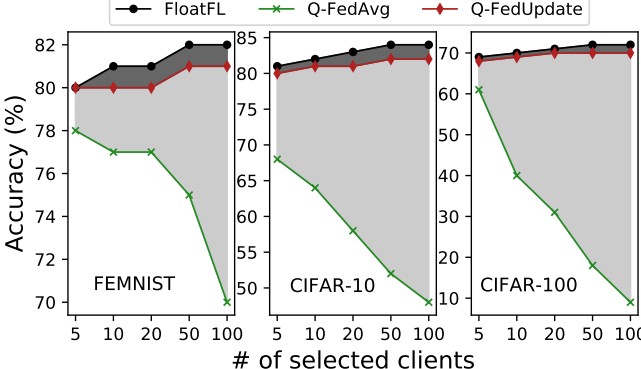

**Figure 8: The scalability (i.e., impacts of more participant devices per round) of Q-FedUpdate and Q-FedAvg. The dark area indicates the gap of convergence accuracy.**

From Figure 8, we observe that Q-FedUpdate greatly improves the scalability to benefit from more participant devices each round compared to Q-FedAvg. In traditional FP32-based FloatFL, more clients participating in each round contributes more diverse training samples, which can accelerate the convergence with higher accuracy. However, this is reversed in Q-FedAvg. More specifically, Q-FedAvg's accuracy drops from 78% to 70%, from 68% to 48%, and from 61% to 9% with the number of selected clients increasing from 5 to 100 on three datasets, respectively. In contrast, Q-FedUpdate achieves a great improvement on convergence accuracy as the increase of selected clients. For example, when $K$=100, Q-FedUpdate can greatly improve the accuracy by 11%, 33% and 60%, respectively. The reason for such performance improvement is that our weight compensation design avoids many effective updates being erased, thus better utilizing the updates from more selected clients.

## 5 RELATED WORK

**FL** is an emerging machine learning paradigm to enable many clients to collaboratively train an ML model while preserving data privacy [21, 30]. This work targets cross-device FL scenario, where each client is a mobile device, and the obtained model can be used to improve user experience on AI-driven mobile Web systems [12, 28, 29]. However, in this scenario, the huge energy consumption of on-device training seriously restricts the large-scale deployment and application of cross-device FL [19]. Many efforts have been invested to accelerate the FL convergence, e.g., model compression to reduce each round's network transmission [17, 33, 34, 36, 40], yet still adopt FP32 format for on-device training.

**Low-precision training** has been studied for many years to reduce the computation overhead on both clouds and edges, e.g., FP16 and INT8 [6, 13, 38, 47, 48]. For example, INT8-based training approaches [38, 47, 48] optimized INT8 training algorithm in both forward and backward passes, which can achieve adequate training speedup with nearly negligible accuracy loss. However, these work mainly focus on a single device's overhead of centralized machine learning tasks, which is not enough for FL training with large-scale mobile devices. We are aware of two similar researches [5, 45] that makes preliminary efforts in introducing low-precision data representation to mitigate device heterogeneity in FL. Our study differs from them in two aspects: (1) We adopt a fully INT8 training algorithm, and accuracy is nearly 5% higher, evaluated on multiple models. (2) We actually deploy the state-of-the-art INT8 training algorithm on CPU/DSP processors of mobile devices, and measure the substantial improvement on clock time, energy consumption.

**Energy optimization for FL.** Many researches focused on how to jointly optimize the computation and wireless transmission energy consumption through resource scheduling [9, 16, 26, 44]. These methods achieved the energy saving by coordinating the computation and communication, as well as the communication resources itself, but ignoring the huge energy consumption of DNN training on mobile devices. AutoFL [20] tailor-designed a reinforcement learning algorithm to judiciously determine the participant devices, which can reduce the number of devices involved in each round. While this method can reduce the average energy consumption of all participant devices, it is more likely to introduce unfairness. This is because some more important devices will often be selected, resulting in excessive energy consumption [24, 25].

## 6 CONCLUSIONS

In this paper, we present a novel energy-efficient FL framework, namely Q-FedUpdate, which enables INT8-based training on energy-friendly mobile DSP chip. It employs an idea of maintaining a global FP32 model where the tiny aggregated model updates can be accumulated, integrated with a CPU-cooperative efficient batch quantization. Compared to existing FL protocols, Q-FedUpdate can both greatly reduce the energy consumption of on-device training and accelerate the model convergence with acceptable accuracy loss.

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
