# OpenReview forum: "Towards Energy-efficient Federated Learning via INT8-based Training on Mobile DSPs"
_ACM.org/TheWebConf/2024/Conference — TheWebConf24_

### Official Review · Reviewer_TCLs · 2023-11-17

**Novelty:** 5
**Technical Quality:** 5

**Review:**

This paper introduces a new energy-efficient FL framework called Q-FedUpdate. This framework allows for INT8-based training on a mobile DSP chip that is designed to be energy-friendly.   The approach involves maintaining a global FP32 model and accumulating small aggregated model updates, which are then integrated with a CPU-cooperative efficient batch quantization.
 In comparison to current FL protocols, Q-FedUpdate offers significant benefits. That effectively minimises power usage during on-device training and speeds up model convergence without compromising precision. The paper's strength is primarily the energy savings claimed in the findings, while its weakness could be the resilience of the model that is not sufficiently mentioned (could be the future direction).

**Questions:**

1. One key question is the relevance of the paper to the conference. Understanding that the current paper is very `packed', it is hard to include additional statements to improve its relevance. The author may explains this issue.

**Ethics Review Description:**

nil

**Reviewer Confidence:**

2: The reviewer is willing to defend the evaluation, but it is likely that the reviewer did not understand parts of the paper

**Scope:**

2: The connection to the Web is incidental, e.g., use of Web data or API

---

### Official Review · Reviewer_SRa1 · 2023-11-20

**Novelty:** 3
**Technical Quality:** 4

**Review:**

This paper addresses a key challenge in FL (federated learning), specifically, high energy consumption associated with cross-device FL training with main focus placed on mobile devices. It starts by introducing and laying out the significance of FL as a privacy preserving ML (machine learning) paradigm and points out the issue of high energy consumption in traditional FL FP32 (32-bit floating-point) operations (mainly with ultra-portable mobile devices). FL is a decentralized ML approach where training occurs across multiple edge devices such as mobile phones whilst enforcing privacy by not sharing data.

The proposed solution in this paper is a novel FL framework known as the Q-FedUpdate. Through the use of INT8 (low-precision 8-bit integer) training on mobile DSPs (digital signal processors), the framework will hopefully maintain a good model accuracy whilst reducing energy consumption significantly. The 3 core innovations in Q-FedUpdate are:

INT8-based training: Q-FedUpdate proposes using INT8 operations for training instead of FP32 as it is less computationally intensive and as a result of such lower energy consumptions.

ECA (error-compensated aggregation): Q-FedUpdate incorporates an error-compensated aggregation protocol that aggregates local model updates (quantization errors) produced by the INT8 training, hence ensuring the model's accuracy is not significantly compromised.

PBQ (pipelined batch quantization): Q-FedUpdate uses a pipelined batch quantization mechanism that allows parallelized CPU-based quantization (FP32 to INT8) and DSP-enabled training. This reduces the computational overhead associated with frequent data quantization such as latency.

**Questions:**

The reviewer has the following questions:

What do you consider as the biggest limitation of Q-FedUpdate and what are your plans for addressing it in future research?

Could you please explain in more detail how the energy consumption reduction was measured in your experiments together with the implications of device operational variations?

Could you please provide us with some insight on the scalability challenges associated with Q-FedUpdate in real-world large-scale FL environments and how it might potentially perform under diverse network conditions?

How dependent is Q-FedUpdate on specific hardware features of mobile DSPs and what are some potential challenges in adaptation to a wider range of mobile devices?

Could you please elaborate on Q-FedUpdate's performance with complex datasets significantly different from CIFAR-10, CIFAR-100, and FEMNIST, especially in real-world scenarios?

How does ECA deal with extreme quantization errors such as those found in complex models or highly variable datasets?

Are there any specific challenges you are expecting or anticipating in the practical deployment of Q-FedUpdate in terms of data privacy and user experience?

A few points that need clarification regarding the proposed method:
Precision limitations with INT8: although ECA was introduced in the paper to address concerns of major quantization errors from FP32 to INT8, it is not known how effective it is when dealing with various complex datasets and models. Furthermore, the tradeoff with accuracy for energy efficiency may cause it to be unsuitable for some ML tasks that demand precision.

Result generalizations: As experiments were only conducted on CIFAR-10, CIFAR-100, FEMNIST, it is unclear how effective the proposed method is with other datasets that have varying complexity.

Hardware differences: It is unclear how well the proposed method will perform on various other mobile DSP architectures (questions adaptability). As Q-FedUpdate is designed with mobile DSPs in mind, its energy efficiency and performance may vary across different hardware. It is likely not optimized for every single mobile device (that have different hardware).

ECA: Although ECA is introduced as an innovative approach for addressing the issues, the paper might not have fully explored the potential challenges/limitations of this method in various different scenarios. The complexity of implementing ECA in real-world large-scale FL environments might also need more investigation.

Energy efficiency: Although Q-FedUpdate is described to be an energy efficiency solution, the paper could potentially go into more detail on how energy consumption is measured at different phases (e.g. model aggregation, data transmission), as well as the operating conditions of the device at that time.

Real-world deployment and scalability: Potential concerns with scalability of Q-FedUpdate in the large-scale FL learning networks of the real-world where it includes numerous other devices, each with their own capabilities. After deployment, there may also be potential concerns with data synchronization, latency in the network, and model update complications/conflicts in a real-world scenario.

**Reviewer Confidence:**

3: The reviewer is confident but not certain that the evaluation is correct

**Scope:**

4: The work is relevant to the Web and to the track, and is of broad interest to the community

---

### Official Review · Reviewer_8tZc · 2023-11-22

**Novelty:** 5
**Technical Quality:** 5

**Review:**

A key challenge in cross-device FL is huge energy consumption. This work performs the first-of-its-kind analysis of low-precision training with an energy-friendly Digital Signal Processor (DSP) on mobile devices for FL. They identify that directly integrating the state-of-the-art INT8 training algorithm and classic FL protocols will significantly degrade the model accuracy. Therefore, they present Q-FedUpdate that aggregates local updates instead of local models. Besides, they propose pipelining the data quantization in CPU and the gradient computation in DSP. Numerous experiments demonstrate the effectiveness of their proposed method.

Strengths:
1.	The problem of reducing the energy consumption in edge devices participating to FL is timely and important.

2.	This paper is easy to follow.

3.	Sufficient experiments are conducted and demonstrate the effectiveness of the proposed method.


Weakness:

1.	There have been works applying model quantization in FL, i.e., [1]. The claim that this is the first work is not accurate. This paper should compare with it.

[1] Bitwidth Heterogeneous Federated Learning with Progressive Weight Dequantization. ICML 2022.


2.	The novelty of aggregating local updates instead of local models is limited. It is straightforward and has been widely adopted in many existing communication works, e.g., in [2].

[2] FedPAQ: A Communication-Efficient Federated Learning Method with Periodic Averaging and Quantization. In Proceedings ofthe International Conference on Artificial Intelligence and Statistics (AISTATS), 2020.

3.	The meaning of the pipeline is not clear. This paper emphasizes in the introduction that reducing the energy is the key objective of this paper. However, the pipeline will not reduce energy consumption but accelerate the training speed. In the meantime, this speedup gain of the pipeline module is unclear and is not presented in the evaluation. Besides, as compared to the communication time which is usually huge, it concerns me if this little performance gain is necessary. This module appears to have few connections to federated learning and seems specifically tailored for on-device learning.

**Questions:**

no

**Reviewer Confidence:**

3: The reviewer is confident but not certain that the evaluation is correct

**Scope:**

4: The work is relevant to the Web and to the track, and is of broad interest to the community

---

### Official Review · Reviewer_f65Y · 2023-11-24

**Novelty:** 5
**Technical Quality:** 5

**Review:**

Summary: This is a very interesting work that focuses on the quantization process in efficient federated learning. The paper finds out the key reason for slow convergence and low accuracy in Q-FedAvg and proposes a simple but effective method to solve the problem. The paper provides detailed analysis and comprehensive experiments to address the problem and show the efficiency of the proposed Q-FedUpdate algorithm.

Strength:
(1)	The paper is well-written and easy to follow. The authors provide enough background information on FL and quantization. The problem of previous methods is clearly explained.
(2)	Q-FedUpdate tries to solve the accuracy decreasing and slow convergence problems caused by quantization. This is crucial for the implementation and widespread adoption of on-device federated learning. The experiment results look good.
(3)	The experiment and analysis part is thorough and robust

Weakness:
(1)	Quantization is also very popular in fine-tuning large-scale foundation models. Will Q-FedUpdate also work on more challenging tasks such as fine-tuning?
(2)	Is Q-FedUpdate able to work together with existing optimizers and schedulers in federated learning? Is it possible to further optimize the performance of Q-FedUpdate by using some optimizers and schedulers?

**Questions:**

please refer to the weakness

**Reviewer Confidence:**

3: The reviewer is confident but not certain that the evaluation is correct

**Scope:**

3: The work is somewhat relevant to the Web and to the track, and is of narrow interest to a sub-community

---

### Official Review · Reviewer_rTqz · 2023-12-03

**Novelty:** 5
**Technical Quality:** 6

**Review:**

Welcome to submit your work to ACM The Web Conference. The proposed Q-FedUpdate is a dedicated FL 8-bit training design for DSP on mobile devices. The proposed design is to address two challenges i.e., performance degradation of combining 8-bit training with conventional FL protocol, and lacks of parallelization for CPU and DSP for 8-bit training. The design is motivated by experimental and theoretical analysis. I appreciate the cross-device FL design and largely enjoy reading the paper, which follows a logical structure to illustrate the motivation, background, core problems and technical contributions of Q-FedUpdate. There are three aspects to applaud in this work:

First, from the technical quality perspective, the design of Q-FedUpdate is well thought, including its protocol, quantization algorithm with error-compensated aggregation, and CPU-DSP quantization pipelining.

Second, from methodology angle, experimental measurements (both Sec 2.2 and Sec 4) and analysis are well executed, providing readers a clear view on why Q-FedUpdate is needed and as well as how Q-FedUpdate performance across models of LeNet-5 and VGG-16.

Third, from system perspective, the scalability analysis in evaluation does provide good insights for potential deployment of Q-FedUpdate in real environment.

Meanwhile, authors shall still explain why the performance of Q-FedAvg drops drastically e.g., what's the reasons behind (because of testing dataset, or selected algorithms compatibility?). I would also prefer to see how much overhead is introduced by Q-FedUpdate e.g., in terms of implementation complexity (open source available?).

**Questions:**

There are two questions for authors to reflect on:

- From the technical angle, will this DSP oriented method also applicable to other accelerators such as TPU and FPGA? If so, can you elaborate how developers or researchers can adopt Q-FedUpdate? The testing with only Xiaomi 10 is relevant but not yet convincing whether the design can perform on other chipset e.g., Apple A-series, M1, or Intel .

- From the originality angle, since the focus is on mobile Web services while Q-FedUpdate is evaluated over handwritten and image datasets, could Q-FedUpdate be applied, in terms of generalization, to other relevant AI-driven applications on top of web? e.g., audio, voice applications. More reflections on this would be useful for the FL and AI communities.

**Reviewer Confidence:**

4: The reviewer is certain that the evaluation is correct and very familiar with the relevant literature

**Scope:**

4: The work is relevant to the Web and to the track, and is of broad interest to the community

---

### Decision · Program_Chairs · 2024-01-22

**Decision:**

Accept

**Comment:**

This paper addresses a key challenge in Federated Learning (FL), specifically, high energy consumption associated with cross-device FL training with main focus placed on mobile devices. The reviewers appreciated the novelty, technical quality, and the energy efficiency of the proposed approach but also had concerns about generalization, limited experimentation, and potential hardware dependencies.

 **Pros:**
 1. Effective Design: The design of Q-FedUpdate is well-thought-out, with effective protocol, quantization algorithm, and CPU-DSP quantization pipelining.
 2. Evaluation: Comprehensive measurements and analysis, providing clear insights into the need and performance of Q-FedUpdate.
 3. Scalability: Potential for real-world deployment and scalability, especially in energy-efficient federated learning on mobile devices.
 4. Energy Efficiency: Significant energy savings without compromising precision, especially relevant for on-device training.

 **Cons:**
 1. Limited Generalization and Experimentation: The testing is somewhat limited (e.g., only on Xiaomi 10), raising questions about generalization across different chipsets and real-world applications.
 2. Unclear Aspects: Some aspects, like the overhead introduced by Q-FedUpdate and the performance gain of the pipeline module, are not clearly addressed.
 3. **Lack of Comparative Analysis**: Needs comparison with similar existing works and clarification on the novelty of aggregating local updates.
 4. Potential Hardware Dependence: Uncertainty about how well the method performs on various mobile DSP architectures.